# Study protocol for the Alzheimer and music therapy study: An RCT to compare the efficacy of music therapy and physical activity on brain plasticity, depressive symptoms, and cognitive decline, in a population with and at risk for Alzheimer's disease

Birthe K. Flo [1]*, Anna Maria Matziorinis [1], Stavros Skouras[1], Tobba Therkildsen Sudmann [2], Christian Gold[3,4,5], Stefan Koelsch[1]*

1 Department of Biological and Medical Psychology, University of Bergen, Bergen, Norway, 2 Department of Health and Function, Western Norway University of Applied Sciences, Bergen, Norway, 3 NORCE Norwegian Research Centre AS, Bergen, Norway, 4 Grieg Academy Department of Music, University of Bergen, Bergen, Norway, 5 Department of Clinical and Health Psychology, University of Vienna, Vienna, Austria

* birthe.flo@uib.no (BKF); stefan.koelsch@uib.no (SK)

## Abstract

### Background

There is anecdotal evidence for beneficial effects of music therapy in patients with Alzheimer's Disease (AD). However, there is a lack of rigorous research investigating this issue. The aim of this study is to evaluate the effects of music therapy and physical activity on brain plasticity, mood, and cognition in a population with AD and at risk for AD.

### Methods

One-hundred and thirty-five participants with memory complaints will be recruited for a parallel, three-arm Randomized Controlled Trial (RCT). Inclusion criteria are a diagnosis of mild (early) AD or mild cognitive impairment (MCI), or memory complaints without other neuro-psychiatric pathology. Participants are randomised into either a music therapy intervention (singing lessons), an active control group (physical activity) or a passive control group (no intervention) for 12 months. The primary outcomes are the brain age gap, measured via magnetic resonance imaging (MRI), and depressive symptoms. Secondary outcomes include cognitive performance, activities of daily living, brain structure (voxel-based morphometry and diffusion tensor imaging), and brain function (resting-state functional MRI).

### Trial status

Screening of participants began in April 2018. A total of 84 participants have been recruited and started intervention, out of which 48 participants have completed 12 months of intervention and post-intervention assessment.

**Data Availability Statement:** This protocol does not report experimental results and data collection is currently ongoing. De-identified data will be made available upon study completion, along with the publication of the results.

**Funding:** A grant by the The Research Council of Norway (RCN; https://www.forskningsradet.no/en/) [Norges Forskningsråd], reference number 260576 was awarded to S.K. The Project is further supported by: Trond-Mohn-Stiftelse (TMS; https://mohnfoundation.no/), Bergens Forskningsstifelse (BFS; https://www.uib.no/foransatte/75175/bergens-forskningsstiftelse), and the Institute of Biological and Medical Psychology (IBMP; https://www.uib.no/ibmp) at the University of Bergen (UiB) in Norway. The funders had no role in study design, data collection and analysis, decision to publish, or preparation of the manuscript.

**Competing interests:** The authors have declared that no competing interests exist.

## Discussion

Addressing the need for rigorous longitudinal data for the effectiveness of music therapy in people with and at risk for developing AD, this trial aims to enhance knowledge regarding cost-effective interventions with potentially high clinical applicability.

## Trial registration

ClinicalTrials.gov identifier: NCT03444181, registered on February 23, 2018.

## Introduction

In 2021, 55 million people worldwide were living with dementia, and this number is expected to double within the next 20 years [1]. Dementia is now the 7th leading cause of mortality globally and among those diseases with the highest costs for society. Alzheimer's disease (AD) is the most common form of dementia accounting for around 60–80% of all cases [2]. AD is a neurodegenerative disease characterized by a progressive impairment of episodic (autobiographical) and semantic memory, disordered cognitive function, a decline in language function, and changes in both affectivity and behaviour [3]. Pharmacological interventions have only limited effects on the symptoms, and to date no cure is available [4]. Therefore, there is a need for non-pharmacological, cost-effective interventions for AD.

There is now consensus that AD begins with a long asymptomatic phase, often referred to as the preclinical stage, where neuropathological changes occur, but cognitive abilities are still unaffected. For example, AD pathology can be detected using prognostic biomarkers more than 10 years before a person reports cognitive changes [5]. The preclinical stage is followed by progressive cognitive decline, referred to as the prodromal stage, prior to impairment and evident AD [6]. Mild Cognitive Impairment (MCI) is considered a prodromal stage of AD. However, progressive neurodegeneration and irreversible cognitive impairment may have already occurred at this stage [7]. Moreover, accumulating evidence indicates that, already before MCI, Subjective Cognitive Decline (SCD) may represent the first symptomatic manifestation of AD [7, 8]. SCD refers to a self-experienced decline of memory and/or other cognitive capacities, without an objective neuropsychological deviation [7, 8].

These findings have led to the notion that the progression of AD be conceptualized along a disease continuum [6], from SCD to MCI to AD, bringing along a shift of scientific interest towards the early identification of individuals at-risk. This interest is also driven by the need to enable clinicians to intervene earlier than presently possible. Research studies have suggested that interventions have a limited effect once irreversible neurodegeneration and cognitive dysfunction has developed [5, 9], and that disease-modifying therapies may be more efficacious in earlier phases of AD. Initiating preventive interventions in the preclinical stage of AD can thus potentially decelerate cognitive impairment and ideally prevent progression to AD. However, pharmacological interventions in individuals with SCD can be problematic due to heterogeneous etiology (only some individuals with SCD convert to MCI, and later to AD), significant side effects [10], and expensive implementation and maintenance. Non-pharmacological interventions could, however, hold promise for the preclinical and prodromal stages of AD.

## Memory and cognition

Mounting evidence indicates that, despite severe impairment of episodic (and moderate impairment of semantic) memory, some patients with AD have nearly preserved memory of musical information, even in the advanced stages of the disease [11, 12]. Moreover, brief music exposure was shown to have positive effects on episodic memory retrieval [13, 14], even when the music was unrelated to the recalled autobiographical event [15], suggesting that musical memory retrieval can lead to benefits in the domains of episodic and semantic memory retrieval.

There is also emerging evidence that music therapy (MT) interventions can improve standardized cognition scores in patients with, or at risk for, dementia. For example, in a study by Särkämö and colleagues [16], 89 patients with mild-to-moderate dementia were randomized to a singing group, a music listening group, or a usual care group, for 10 weeks. Compared with usual care, both singing and music listening improved orientation, attention, executive function, and general cognition. Singing also enhanced working memory and caregiver well-being. In addition, Gómez Gallego & Gómez García [17] found a significant increase in MMSE scores in participants with mild-to-moderate AD after 6 weeks with MT. Music listening for 3 months has further been shown to significantly enhance both subjective memory function and objective cognitive performance in participants with SCD [18]. Thus, MT interventions might offer promising results for improved outcomes in the preclinical and prodromal stages of AD.

A recent random-effects meta-analysis of 110 MT studies with dementia patients reported beneficial effects of active MT on global cognition [19]. Notably, it was concluded that there is a need for randomized controlled trials (RCTs) with larger sample sizes to better elucidate the impact of MT on cognitive functions in dementia patients. This conclusion echoes those of previous reviews finding mixed results, weak effect sizes, and a lack of high-quality longitudinal MT studies in dementia patients [4, 20]. Note that these reviews [4, 19, 20] dealt with dementia in general (not AD in specific), and to the best of our knowledge there is lack of published RCTs on the effects of MT on cognitive function in AD (and lack of published RCTs on the effects of MT on brain structure and brain function).

## Behavioural and psychological symptoms

Music can reduce behavioural and psychological symptoms, such as depression and anxiety in AD patients. A systematic review [21] summarized the results of 38 systematic reviews on various non-pharmacological interventions and found that music interventions were best at reducing behavioural symptoms, specifically agitation and anxiety, in dementia. An RCT showed that patients with mild to moderate AD had a reduction in anxiety and depression scores over the course of 16 weeks with music listening (receptive MT) [22]. A later review found that MT was effective in reducing depressive and overall behavioural symptoms in dementia [4]. Results from a study that collected AD patients' saliva before and after MT to quantify cortisol levels indicated that music could lower overall stress levels [23].

## Brain plasticity

The strong effects of music, in particular music making, on brain plasticity have been shown in several neuroscientific studies with both healthy individuals [e.g., 24–26], and patients [e.g., 24, 27, 28]. Furthermore, a study identified a brain area involved in musical memory retrieval (the pre-supplementary motor area) and found that this area was among the last to show atrophy in AD patients [29]. In an functional Magnetic Resonance Imaging (fMRI) study, 10 AD

patients received 6 months of singing intervention with karaoke [30]. The results suggested a decreased parietal activation in an fMRI karaoke task, indicating enhanced neural efficiency of cognitive processing. However, there is a lack of studies investigating effects of MT interventions on brain plasticity in AD patients.

## Music therapy techniques

A wide and heterogenous range of MT techniques can be observed in the literature. This heterogeneity may affect the results of different studies. For example, active MT techniques (music making), but not receptive techniques (music listening), appear to have beneficial effects on global cognition in dementia patients [19]. Active music making in the form of singing (as planned in the present study) has been applied in a few studies, and a systematic review by McDermott et al. [20] found that singing featured as an important medium for change in dementia patients. Individualized music (considering participants' preferences) [17, 22, 31] and familiar, rather than unfamiliar, music [32] also seems to have more beneficial effects.

In summary, music has powerful effects on memory in patients with neurodegenerative diseases. However, although there is suggestive evidence for beneficial effects of active music interventions in patients with AD, rigorous longitudinal research on this subject is lacking. Systematic reviews [4, 19, 20, 33–35] have continued to highlight the methodological limitations in this research field, emphasising an insufficient number of studies, small sample sizes, scarceness of RCTs and longitudinal study designs, and a shortage of studies with active control groups (instead of simply using "treatment as usual" as a comparison group). In addition, there is a lack of RCTs on the effects of MT in AD using Magnetic Resonance Imaging (MRI) measurements. Therefore, the aim of the present study is to provide rigorous longitudinal data on the effects of music therapy on brain plasticity, mood, and cognition in a population with AD and at risk for AD. In a randomized controlled intervention trial, a sample of participants with AD or at risk for AD, without any other neuropsychiatric pathology, will undergo twelve months of active MT consisting of singing lessons and choir singing. Structural and functional MRI will be used to determine changes in the brain age gap (as compared to two control groups), integrity of fiber tracts, and resting-state brain activity. The study will include an active control arm, involving weekly sessions of Physical Activity (PA), to better characterize the specific effects of MT and to evaluate the effects of PA as a beneficial intervention for AD. PA was chosen as an active control condition because it has been shown to be a promising non-pharmacological intervention for dementia [33]. One study found protective effects of PA on hippocampal atrophy in elders at genetic risk for AD [36], and a meta-analysis suggested that PA was associated with a reduced risk of Alzheimer's disease in adults over the age of 65 years [37].

## Objectives

The main objective of our study is to evaluate the efficiency of music therapy and physical activity in AD prevention through a longitudinal RCT design with neuroimaging. We hypothesize that MT can (a) decelerate brain ageing, (b) ameliorate depressive symptoms, (c) decelerate or ameliorate cognitive decline, (d) increase the quality of life for both patients and family members, (e) prolong the time patients can stay out of nursing homes. Moreover, we aim at investigating which cognitive, emotional, and social factors drive effects of MT and PT on brain atrophy, mood, and activities of daily living.

## Methods/Design

### Study design

The study is a parallel, three-arm RCT. The trial is conducted in Bergen, Norway (a city with a population of around 300.000 inhabitants). The schedule of enrolment, interventions, and assessments can be seen in Fig 1. Participants are being randomised to either a music intervention (singing lessons), a non-musical intervention (physical activity) or no intervention (passive control group) for 12 months including weekly intervention sessions. Block randomisation with randomly varying block sizes of 3 or 6 is being used to ensure balance as well as unpredictability. The randomisation list is computer-generated and concealed by a researcher (CG) who has no direct contact with the participants. The randomisation results are revealed to the clinical investigators only after inclusion is confirmed (eligibility confirmed, informed consent signed, and baseline assessment completed). Blinding is difficult to accomplish. Participants and therapists cannot be blinded to the intervention they receive or provide. Blinding of assessors is also difficult, as they are often the ones who oversee data collection and attendance, as well as are in contact with the participants. An overview of the study design is shown in Fig 2.

| | STUDY PERIOD | | | | |
| --- | --- | --- | --- | --- | --- |
| | Enrolment | Allocation | Post-allocation | | Close-out |
| TIMEPOINT** | *-1d* | *0* | *1d* | *12m* | *t$_x$* |
| **ENROLMENT:** | | | | | |
| Eligibility screen | x | | | | |
| Informed consent | x | | | | |
| Allocation | | x | | | |
| **INTERVENTIONS:** | | | | | |
| Music Therapy | | | ←——————→ | | |
| Physical Activity | | | ←——————→ | | |
| No intervention | | | ←——————→ | | |
| **ASSESSMENTS:** | | | | | |
| **Baseline variables:** Age, gender, medication, education, previous ailments, smoking, DRS-15, Gold MSI. | x | | | x | |
| **Primary outcome variables:** BrainAGE (MRI) and GDS | x | | | x | |
| **Secondary outcome variables:** MMSE, FCSRT, Wordlist, SCD-Q, CWIT, ADL, PROMS, grip strength, SPPB, walk test, IPAQ, physical activities, DTI and rs-fMRI. | x | | | x | |

**Fig 1. The schedule of enrolment, interventions, and assessments.** Abbreviations: DRS-15 –Dispositional Resilience Scale 15; GOLD MSI—Goldsmiths Musical Sophistication Index; BrainAGE—Brain Age Gap Estimation; MRI—Magnetic Resonance Imaging; GDS—Geriatric Depression Scale; MMSE—Mini Mental Status Examination; FCSRT—Free and Cued Selective Reminding Test; SCD-Q—Subjective Cognitive Decline-Questionnaire; CWIT—Colour Word Interference Test; ADL—Activities of Daily Living; PROMS—Profile of Music Perception Skills; SPPB—Short Physical Performance Battery; IPAQ—International Physical Activity Questionnaire; DTI—Diffusion Tensor Imaging; rs-fMRI—resting state functional MRI.

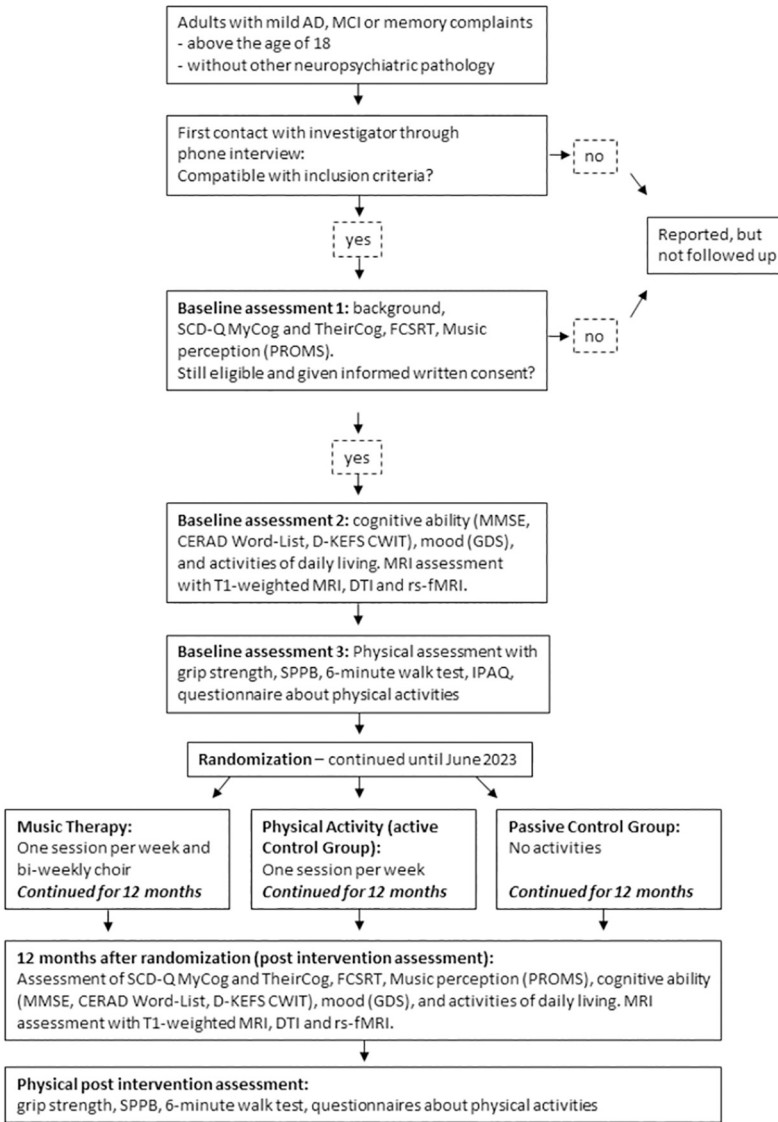

**Fig 2. Flow chart of the study design.** Abbreviations: AD—Alzheimer's Disease; MCI—Mild Cognitive Impairment; SCD-Q—Subjective Cognitive Decline-Questionnaire; FCSRT—Free and Cued Selective Reminding Test; PROMS—Profile of Music Perception Skills; MMSE—Mini Mental Status Examination; CWIT—Colour Word Interference Test; GDS—Geriatric Depression Scale; MRI—Magnetic Resonance Imaging; DTI—Diffusion Tensor Imaging; rs-fMRI—resting state functional MRI; SPPB—Short Physical Performance Battery; IPAQ—International Physical Activity Questionnaire.

## Participants

Trial participants are adults with mild (early) AD, MCI or memory complaints recruited through advertisements and collaborating institutions (departments at Haukeland University Hospital, Haraldsplass Diaconal Hospital, Olaviken gerontopsychiatric hospital, and Bergen municipality) who comply with the following criteria:

**Inclusion criteria.** Adults with a diagnosis of mild AD or MCI due to AD, or without a diagnosis but with memory complaints without any other major neuropsychiatric pathology are eligible to participate. Our criteria for mild AD are: (i) having a higher score than 17 on the Mini Mental State Examination (MMSE), (ii) still living at home (not in a nursing home), and

**Table 1. Stages of Objective Memory Impairment (SOMI) defined by free recall and total recall score ranges and years to diagnosis [39].**

| SOMI | Free Recall (FR) scores | Total Recall (TR) scores | Years to diagnosis: Mean (SD) | Class of Memory Impairment |
|---|---|---|---|---|
| 0 No Memory Impairment | >30 | >46 | 7.05 (2.80) | None detected by FCSRT |
| 1 Subtle Retrieval Impairment | 25–30 | >46 | 4.89 (2.48) | Free recall declines at a constant rate. Storage is preserved. |
| 2 Moderate Retrieval Impairment | 20–24 | >46 | 4.03 (2.62) | Rate of free recall decline doubles. Executive dysfunction accelerates. Storage is preserved. |
| 3 Subtle Storage Impairment | any | 45–46 | 2.09 (1.91) | Cuing fails to normalize total recall. |
| 4 Significant Storage Impairment compatible with dementia | any | 33–44 | 0.86 (1.30) | Intellectual decline accelerated heralding ADL impairment |
| 5 Moderate Episodic Memory Impairment | any | ≤32 | | SOMI was not intended to measure impairment of moderate severity, but stage 5 was added to accommodate these participants |

(iii) being able to provide informed consent. Adults without a clinical diagnosis are eligible if they experience memory complaints (ranging from subjective complaints to objective measurable impairments), as long as they do not meet the exclusion criteria. Participants also need to be able to complete questionnaires in Norwegian, undergo MRI, and attend assessments and weekly intervention in the Bergen area in Norway. All randomised participants will be retained and analysed according to the intention-to-treat principle.

**Exclusion criteria.** Exclusion criteria for the study are (1) other Dementia types (i.e. Lewy body dementia, Frontotemporal Dementia, Vascular Dementia), (2) vascular disorders (history of heart disease, heart attack, heart surgery, stroke), (3) traumatic brain injury, (4) neurological illness (i.e. Multiple Sclerosis, Epilepsy), (5) severe auditory impairments, (6) physical immobility, (7) severe psychiatric disorders (Major Depressive Disorder, Bipolar Disorder, Schizophrenia, Psychotic Symptoms), (8) metal in the soft tissue of the body (i.e. pacemakers), (9) moderate or severe Alzheimer's Disease (MMSE score < 17).

These criteria resulted in a sample of participants with memory performance that ranges from marginal subtle memory impairment to early-stage dementia. To further subcategorize the participants, the Stages of Objective Memory Impairment (SOMI) framework is used [38, 39]. The SOMI framework provides a neuropathologically validated staging system for episodic memory impairment in the AD continuum and identifies four predementia stages and two dementia stages. SOMI was developed based on literature mapping of performance in the Free and Cued Selective Reminding Test (FCSRT) [40, 41] to clinical outcomes and to biomarkers in longitudinal aging studies. Table 1 is derived from Grober and colleagues [39] and shows the SOMI framework defined by Free Recall and Total Recall score ranges and the estimated time to clinical dementia at each of the stages. The estimated time to diagnosis has been validated by analyzing the SOMI scores of 142 incident AD cases tested annually for up to 10 years [38].

## Power calculation and sample size

Power was calculated for detecting differences in the structural brain data (difference between brain-AGE pre and post intervention). The effect size to be expected is uncertain but is assumed to be in the medium-to large range (i.e., Cohen's d between 0.5 and 0.8). We aim to achieve 80% power in two one-sided t-tests, Bonferroni-adjusted for two comparisons of interest (music therapy vs. passive control, physical activity vs. passive control, i.e., the one-sided significance level will be 5%/2 = 2.5%). Power calculations (using G* Power 3) indicate that a valid sample size of n = 35 per group will result in 80% statistical power if the effect is d = 0.68,

which is in the medium-to-large range. To account for attrition, we aim to recruit 40 to 45 participants per group for a total of 135 study participants across the three groups.

## Interventions

Participants will be randomly assigned to one of the following three conditions, and will continue to receive what is prescribed to them according to the randomisation:

1. Music therapy: The music intervention consists of weekly individual singing lessons and biweekly choir practice for 12 months (around 40 sessions considering holidays and public holidays). Participants are provided with audio files of the songs and are asked to practice daily with the help of these audio files for 30 minutes per day.

2. Physical activity: The physical activity intervention entails weekly group training sessions for 12 months (around 40 sessions). Participants are also encouraged to be active in-between sessions.

3. Passive control group: The passive control group does not receive any intervention for 12 months. Participants continue their lives as usual and are asked not to participate in other concurrent research studies during these 12 months. After the participants have completed the post-intervention assessments, they are offered to join the choir or the physical activity group.

**Description of music therapy.**    Participants randomized to the MT group participate in an individual MT setting as well as in a choir. The duration of the weekly individual singing lessons is 45–60 minutes. Similarly, the biweekly choir sessions have a duration of around 45 minutes. The MT is provided by a music therapist, or advanced MT students, or psychology students with a musical background, or psychologists with a musical background, under the supervision of a music therapist. Participants do not need prior musical knowledge or experience, and the sessions are individually adapted. The individual MT sessions include warm up, singing training, rehearsal of songs for the choir, and end with music listening. The repertoire consists of both known and unknown songs, in addition to various forms of canon and polyphonic singing. The songs are selected according to different degrees of difficulty, so that all participants will experience challenges adapted to their level. The choir is intended as a social meeting place where the joy of singing, and sense of accomplishment are as important as rehearsing a new repertoire. The choir is divided into three main parts: warm-up, rehearsal of previous songs and new material.

**Description of physical activity.**    Participants randomized to the Physical Activity (PA) group attend weekly group sessions of 70–90 minutes. Instructors are physiotherapists (PTs) and sports scientists. The PA sessions are designed based on principles from high intensity interval training (HIIT) to affect physical as well as social outcomes, strength, endurance, coordination, co-operation and attention, while maintaining a positive emotional experience. PA-sessions are held outdoors during spring and early autumn, and indoors in late autumn and winter. Activities include warming up, individual and group-based activities (competitions) with varied intensity levels and cooling down. Every session includes, but is not limited to, activities with gross- and fine motor movements, coordination between hands, fingers and feet, upper and lower body, crossing the body's midline, balance, walking, standing, jumping, and running. The aim of the sessions is identical, but the choice of activities varies. The instructors are free to choose from a wide variety of equipment and props. Participants' reactions are observed in every session in accordance with the Borg Rating of Perceived Exertion

[42] (e.g., observing breathing, face colour, movement quality, balance, attention, and reaction time). Participants are encouraged to give feedback and to suggest activities. In the interest of feasibility, we did not include the assessment of metabolic parameters, nor pulse measurements during the physical activity intervention, but future studies might include such measures as they could provide further insights into their role in mediating pro-cognitive effects.

**Assessment of treatment fidelity.** Attendance rate is documented by the therapists in charge of the intervention. The therapists also document significant events. Participants who have not been present repeatedly are contacted to monitor the reasons for their absence and to encourage attendance. If someone drops out of MT or PA, they are still asked to complete the post-intervention assessment after 12 months. Information on concomitant care will be recorded both before randomization and post intervention.

**Assessment and outcome measures.** Prior to the baseline assessment, participants undergo a telephone interview which reviews information regarding the inclusion and exclusion criteria. Participants who are not eligible are documented but not included in the study and not followed up. When included, a comprehensive testing protocol is applied, including cognitive, psychological, physical, and musical measures, as well as MRI, Diffusion Tensor Imaging (DTI) and functional MRI (see section Outcome Measures for details). Participants undergo this testing protocol at baseline and at 12-month follow up (i.e., before and after the one-year interval of the interventions or the passive-control condition). Assessors are not aware of participant's group allocation during baseline assessment. Assessment is split into three sessions to avoid fatigue during testing. During the first session, baseline characteristics are assessed for describing the sample, including age, gender, medication, education, previous ailments, smoking history, preferred intervention, resilience (using the revised Norwegian Dispositional Resilience Scale 15) [43], and self-reported musical skills (using two subscales, active engagement and musical training, of the Goldsmiths Musical Sophistication) [44], and neuropsychological assessment is performed. Only if the results of these tests are within the inclusion / exclusion criteria, participants are admitted to the second baseline assessment session, in which neuropsychological assessment is continued and MRI measurements are performed. The third session comprises a physical fitness assessment, administered by registered physiotherapists and sports educated personnel. An overview of the testing sessions and the study design is shown in Fig 2.

**Primary outcomes.** *Brain Age*. The primary outcome of the study is to examine brain plasticity, specifically to identify beneficial effects of MT on brain age. The machine learning algorithm known as Brain Age Gap Estimation (BrainAGE) [45] will be used to estimate participants brain age gap. BrainAGE is based on structural T1-weighted MRI scans (we use a GE Discovery MR750 3 Tesla (3T) MRI scanner). Gaser et al. [46] found that BrainAGE, with accuracy rates of up to 81%, outperformed cognitive scales and CSF biomarkers in predicting conversion of MCI to AD within 3 years of follow-up. It will be examined whether differences in BrainAGE between pre- and post-intervention differ between groups. In specific, we will test for a slower progression of the brain age gap in the MT group compared to the control groups.

*Depressive symptoms*. It is expected that depressive symptomatology will be lower after the intervention, compared to before the intervention, for participants in both active interventions (MT and PA), compared to the passive control group. Depressive symptoms are measured using the Geriatric Depression Scale (GDS) [47]. The GDS total score can range from 0 to 30. Higher scores indicate more severe depressive symptoms. A score of > 11 indicates depression, with a sensitivity of 84% and specificity of 95% [47]. However, for patients diagnosed with dementia, sensitivity and specificity is somewhat lower regardless of the cut-off score [48].

**Secondary outcomes.** *Cognitive measures.* Secondary outcomes include memory measurements, to assess whether memory disturbances have progressed, receded, or stabilized over the course of 12 months across the three intervention groups.

- The *Mini-Mental State Examination* (MMSE) [49] is the most frequently used screening tool for dementia [50]. This study utilizes the MMSE-Norwegian Revised version (MMSE-NR) [51], which is a brief global assessment of cognitive status with 30 items to examine orientation, attention, memory, language, and visuo-spatial skills. A full score of 30 indicates normal cognition, while an MMSE score <17 is often reported as moderate-severe cognitive impairment [52, 53]. It has been advised that MMSE is insufficient for detecting subtle changes of cognition in the preclinical and prodromal stages of AD [54].

- The *Free and Cued Selective Reminding Test* (FCSRT) [40, 41] has been proven valuable in the characterization of the memory impairment across the AD spectrum [55]. The International Working Group on Alzheimer's disease (AD) have recommended the FCSRT to assess episodic memory [56], because it shows high sensitivity and specificity in differencing AD patients from healthy controls, AD from non-AD dementias [57], and patients with probable prodromal AD from subjects with MCI or SCD that do not progress to AD dementia [58–60]. The test procedure comprises of two phases: a study phase and a test phase. During the study phase, participants are asked to identify 16 pictures, each from a different semantic category. Immediate cued recall is tested by prompting with the category cue, and participants are reminded of items they do not recall. During the Test phase, three trials of free and cued recall are administered. The scores calculated from this are total Free Recall (FR; the cumulative sum of free recall from the three trials; 0–48), Total Recall (TR; the cumulative sum of free recall + cued recall; 0–48), and cue efficiency (total recall—free recall).

- *Verbal memory* is evaluated using the Word List Memory test from The Consortium to Establish a Registry for Alzheimer's Disease (CERAD) [61]. One of the most characteristic aspects of memory problems in AD is the rapid forgetting of newly learned verbal information [62]. The CERAD Word List Test assesses verbal learning (by visually presenting 10 unrelated words across three trials, with a score ranging from 0 to a maximum score of 30), delayed free recall (total score ranging from 0 to 10) and recognition (by mixing in 10 new words, with the total score ranging from 0 to 20).

- The *Subjective Cognitive Decline Questionnaire* (SCD-Q) [63] is used to measure self-perceived cognitive decline. SCD-Q has two parts, MyCog is answered by the subject and TheirCog is answered by the informant or next of kin. It contains 24 items about perceived decline in memory, language, and executive functions in the last two years. Total score of both parts of SCD-Q range from 0–24, with higher ratings indicating greater perceived cognitive change. A score ≥ 7 on both MyCog and TheirCog indicates SCD [63].

- To measure changes in executive functions, the Delis-Kaplan Executive Function System's (D-KEFS) [64] version of the Stroop task, called the *Color Word Interference Test* (CWIT) is administered. The CWIT, which consists of four conditions (color naming, word reading, inhibition, and inhibition/switching), uses age-corrected scaled scores that have a Mean of 10 and a Standard Deviation of ± 3.

**Behavioural measures.**

- The Lawton *Activities of Daily Living* (ADL) scale [65] is used to assess participants' independence. The ADL has two scales: Instrumental ADL (IADL) is the ability to perform tasks

such as using a telephone, doing chores, and handling finances. It includes 8 items with a maximum score of 31. The personal ADL scale (PADL) assesses the ability to perform personal tasks, e.g., maintain hygiene, and consists of 6 items with a maximum score of 30. Low scores indicate high independence, while high scores indicate low independence.

**Music psychology measures.**

- The Mini-version of the *Profile of Perception of Music Skills* (Mini-PROMS) [66] is employed for an overall assessment of perceptual musical ability. The PROMS is a standardized instrument to measure musical competence [67]. The Mini-PROMS is an online assessment that takes approximately 15 minutes to complete, measuring discrimination skills for timbre, tuning accuracy, and timing-related skills. It consists of four subtests: melody, tuning, tempo and accent (the relative emphasis given to certain notes in a rhythmic pattern). Each subtest has 8–10 trials, in which participants evaluate whether a reference stimulus and a probe stimulus are the same. Difficulty levels are engineered by decreasing the differences between reference and probe stimuli.

*Physical ability measures.* Physical fitness assessments consist of a six-minute walk test [68], a grip strength test [69], a qualitative questionnaire about everyday physical activities, as well as the following:

- The *Short Physical Performance Battery* (SPPB) [70], that is comprised of a timed sit-to-stand test, a four-meter walk test and balance tests, with a total score of 0–12, where a higher score indicates better physical performance.

- *The International Physical Activity Questionnaire* (IPAQ) short version [71], that consists of 7 open-ended questions surrounding individuals' physical activity in the last 7 days.

- A computerized *Finger Tapping Test* (FTT), that is administered to examine motor functioning, specifically motor speed, and lateralized coordination.

*Caregiver assessment.* In addition to the assessment of the participants, psychometric and quality of life measures from family caregivers of participants with a diagnosis of MCI or AD is also obtained. Beck's Depression Inventory (BDI-II) [72] is a 21-item self-report instrument intended to assess the presence and severity of depressive symptoms. Each item is answered on a four-point scale, with a total score ranging from 0 to 63 points, where higher scores indicate greater depression severity. The Burden Scale for Family Caregivers (BSFC) [73] is a self-report scale with 28 items for measuring caregiver burden. The total score ranges from 0 to 84 points, where higher scores indicate greater caregiver burden.

*Brain Structure.* MRI and DTI are used to examine the effects of MT and PA on grey matter volume and white matter structural integrity, respectively. Individuals with MCI show both grey and white matter changes, as well as fiber tract abnormalities within the medial temporoparietal network associated with episodic memory impairment [74]. These regions also show the strongest predictive value for the discrimination between MCI/AD patients and healthy control subjects [74]. We will use voxel-based morphometry (VBM) to determine which brain regions are associated with social and cognitive factors, in order to investigate the beneficial effects of MT on brain atrophy. DTI will be used to assess integrity of fiber tracts by analysing measures such as mean diffusivity and fractional anisotropy, as well as streamline density volume.

*Brain Function.* Resting-state functional MRI (fMRI) activity begins to decline during the preclinical stage of AD, drops substantially in MCI due to AD, and declines further during the

subsequent stages of the disease [74]. Moreover, reduced default mode network connectivity during resting-state fMRI is considered predictive of impending clinical progression to dementia [74]. Functional connectomics methods, including Eigenvector Centrality Mapping, will be used to investigate the effects of MT and PA on global brain function and the default mode network during a 7-minutes resting-state scan and a 7-minutes passive music listening scan (previous studies indicate that the default mode network is activated during rest, as well as during passive music listening) [75]. Participants choose a genre (all instrumental pop, rock, classical, jazz, world, or folk music) to listen to during the music-listening fMRI session.

## Protocol amendments

The current Study Protocol is version number 1.6, as the study design has gone through several modifications for both practical and scientific reasons. A retrospective feasibility assessment was conducted, and the results generated recommendations that were concurrently implemented to the study design [76]. Changes to the protocol compared with its original published version are shown in S1 Table.

## Statistical analyses

To adhere with RCT requirements, primary analysis will be conducted in the 'intention to treat population', i.e., all who were randomised and for whom outcome data are available will be analysed in the group to which they were randomised, regardless of whether they received the full intended intervention. In case of incidental findings of a non-AD severe disease, such participants will be excluded from the final statistical modelling. Data will be evaluated with regards to meeting parametric assumptions, and appropriate parametric as well as non-parametric tests will be performed accordingly with the objective of assessing the differential effects of MT and PT on the primary and secondary outcome measures. Tests will be one-sided and will be Bonferroni-adjusted for the two comparisons of interest, as described in Power Calculation and Sample Size. In addition, we will model medication effects in a statistical analysis, to investigate the potential additive effect of the interventions combined with medications, such as antidepressants, in order to control for potential confounding factors.

## Ethics

The study was registered on ClinicalTrials.gov on February 23, 2018, with the identifier NCT03444181. The Regional Committee for Medical and Health Research Ethics has approved the study (reference number 2018/206). Written informed consent is obtained, after fully explaining the study and answering all questions the participant might have, as well as making sure the participant know that they are free to withdraw at any time without repercussion. Written informed consent is also provided by the caregivers. The data are de-identified, and the link between the participant ID number and identifiable information are restricted to authorized investigators. This information is stored in a locked filing cabinet and will be destroyed upon study completion. Neuroimaging data are anonymized on two levels: (1) No identifiable information is charted in the MRI scanner, and (2) an automated computer program pre-processes the anatomical images to remove any information that can be used to reconstruct participants faces. Paper records (questionnaires and tests) will be registered in an electronic spreadsheet, verified, and later destroyed. All data from the study will be stored and processed on an encrypted server. All investigators will have access to the final, de-identified, dataset. The MRI scanning is non-invasive, and not considered to pose any health risks for the participants. Random allocation of participants to intervention groups is considered reasonable because no adverse effects are expected in any of the interventions. Participants allocated

**Table 2. Means and standard deviations for the participants in the different SOMI stages.**

| Measures | SOMI 0 (*n* 21) | | SOMI 1 (*n* 16) | | SOMI 2 (*n* 14) | | SOMI 3 (*n* 5) | | SOMI 4 (*n* 13) | | SOMI 5 (*n* 14) | |
|---|---|---|---|---|---|---|---|---|---|---|---|---|
| | *M* | *SD* | *M* | *SD* | *M* | *SD* | *M* | *SD* | *M* | *SD* | *M* | *SD* |
| Age | 65 | 9.8 | 69.8 | 10.6 | 71.6 | 9.4 | 77 | 6.6 | 76.5 | 7.2 | 73.9 | 4.7 |
| Gender (m = 1) | 1.5 | .5 | 1.4 | .5 | 1.5 | .5 | 1.8 | .4 | 1.3 | .5 | 1.6 | .5 |
| FCSRT FR | 34 | 2.5 | 27.8 | 1.7 | 20.4 | 2.3 | 19 | 3.9 | 15.2 | 8.4 | 1.3 | 1.3 |
| FCSRT TR | 47.9 | .3 | 47.6 | .5 | 47.9 | .4 | 45.8 | .4 | 40.8 | 4.4 | 15.8 | 10.3 |
| SCD MyCog | 12.5 | 4.2 | 12.9 | 5.1 | 14 | 10.5 | 13.2 | 5.5 | 8 | 6.2 | 11 | 4.2 |
| SCD TheirCog | 8.4 | 4.1 | 11.1 | 5.1 | 3.4 | 3.5 | 17 | 6.1 | 12 | 7.2 | 16.5 | 3.5 |
| MMSE | 28.4 | 1.7 | 28.3 | 2.2 | 27.2 | 1.9 | 25.4 | 2.7 | 24.2 | 3 | 18.9 | 2.6 |
| BrainAGE | -.5 | 3.6 | -3.3 | 8.3 | 1.9 | 5.7 | 5.9 | 4.8 | 4.4 | 7.1 | 7.9 | 8.8 |

Note: SOMI stage 4 and 5 includes the earliest participants in the study, with a diagnosis of AD, and are therefore missing SCD-Q and FCSRT scores. They are classified into either stage 4 or 5 based on whether their MMSE score is above or below 25 points [77].

to the passive control group will be able to participate in the choir or physical activity group after their post-intervention measurements. A data monitoring committee is not needed because the interventions have no known side effects and the study it is a small, single site trial. If brain abnormalities, severe memory problems, or elevated depression scores are discovered, the participant will be informed and encouraged to get a clinical examination.

## Dissemination plan

The plan for dissemination and communication of results is based on several pillars:

(1) Publication of scholarly articles in peer-reviewed international scientific journals, (2) conferences, conference posters, and conference proceedings, (3) communication to popular audiences and patient communities, and (4) communication of results to music therapists in the Bergen municipality (who is involved as a partner in the project), as well as to both music therapists and neurologists beyond Bergen (nationally and internationally, with the help of the network of national and international project partners from Norway, Germany, Austria, the UK, and the USA).

## Trial status

Since the beginning of the recruitment in April 2018 and until today, 84 participants have completed the baseline assessment and been randomized. Table 2 shows the means and standard deviations for some of the baseline measurements for the participants in the different SOMI stages. So far, 48 participants have had a post-intervention assessment (11 from PA, 18 from MT, and 19 from CG). The dropout rate is 25%, where 21 participants who have been randomized have dropped out. A chi-square test of independence showed that there is no significant association between SOMI stage and group allocation X2 (10, N = 83) = 6.26, *p* = .79. Table 3 shows how many from each SOMI stage are in the different intervention groups. More participants will be recruited through 2022 until the target sample size of 105 participants with pre-and post-tests is met. The final data collection sessions are expected to take place in June 2023. Data entry and processing are ongoing.

## Discussion

Although there are indications that MT yields beneficial effects for AD patients, the literature consists of mixed results, weak effect sizes, and a lack of longitudinal RCTs [4, 20]. The aim of

Table 3. Group allocation of SOMI stages.

| SOMI stage | Group Allocation | | | Total |
|---|---|---|---|---|
| | Music Therapy | Physical Activity | Passive Control | |
| Stage 0 | 6 | 7 | 8 | 21 |
| Stage 1 | 4 | 7 | 5 | 16 |
| Stage 2 | 8 | 4 | 2 | 14 |
| Stage 3 | 2 | 1 | 2 | 5 |
| Stage 4 | 4 | 5 | 4 | 13 |
| Stage 5 | 4 | 4 | 6 | 14 |
| Total | 28 | 28 | 27 | 83 |

the present study is to provide rigorous longitudinal data on the effects of MT and PA on brain plasticity, mood, and cognition in a population with, and at risk for, AD. Some of the strengths of this study are: (1) The study is an RCT with an adequate sample size for the detection of statistical differences of medium-to-large effect size; (2) The use of an active control arm (PA), as well as a passive control arm ensures experimental rigour with regards to group comparisons. Moreover, the potentially beneficial effects of general social engagement are balanced between the MT and PT interventions; thus, any variance associated with social engagement can be controlled for in group comparisons; (3) The active control arm also enables us to assess the effectiveness of physical activity for people with, and at risk for, AD; (4) By incorporating MRI measurements it is possible to explore the neural correlates of the MT and PT interventions.

An intervention period of 12 months provides a valuable opportunity of looking at more long-term effects. However, 12 months of weekly sessions require considerable time and effort from the participants and thus attrition is expected to be a problem. Preliminary analyses show a dropout-rate of 25%. The reasons for low attendance and dropouts are being monitored to account for attrition bias.

Investigating a sample of participants with memory performance that ranges across the AD-spectrum, from marginal, subtle memory impairment to dementia, has both practical and scientific advantages. Interventions at a stage before progressive neurodegeneration has occurred might lead to a higher compliance rate and lower attrition due to the progressive nature of the later stages of the disease. Furthermore, it is likely that early interventions have greater chances for a deceleration of cognitive decline, and perhaps even preventing the onset of, AD. Note that, in the absence of cerebrospinal biomarker data, one cannot be certain that the participants with memory impairments (based on SOMI staging) who lack a clinical diagnosis, express such impairments due to AD, or are at risk for progressing to AD. In our study, we attempt to account for this in part by excluding people with other known diseases that can cause memory impairments (e.g., vascular diseases and psychiatric disorders).

The COVID-19 global pandemic has affected both the recruitment and the data quality in the study. With a vulnerable population and enforced lockdown measures during 2020 and 2021, the recruitment, the testing, as well as the interventions had to be put on hold multiple times. Although the therapists tried to accommodate for the lockdowns by offering sessions online or over the telephone, this change and pause in the interventions as well as delayed post-intervention assessments are expected to lower the data quality. Furthermore, a suggested link between COVID-19 and the acceleration of cognitive deterioration (1), social isolation being strongly intertwined with depressive symptoms in the general population of older adults [78], and limited social interactions being a risk factor for AD [79], the pandemic is expected to influence the results of the study. Nevertheless, any positive effects of MT and PA on brain

aging and depressive symptomatology, that may be observed, would be expected to be generalisable to the conditions of normal life (i.e., in the absence of a pandemic and the associated measures). In addition, we will attempt to include attendance rates and other variables influenced by the pandemic, as regressors in our data analysis.

The present study has a high clinical applicability, as the interventions can be transferred and applied within clinical settings. Conclusions from this study can contribute to providing evidence-based treatments for the prevention of memory decline in people with increased risk for AD. The study has a partnership with the Bergen municipality, that is involved in an advisory role with an interest in making MT and PA more accessible for the elderly in the local community. The municipality will also help to disseminate the study's findings to relevant patient organizations, institutions, and music therapists. MT is already part of the municipality care system, and positive results from the study would support expanding the provided MT services and setting MT in place in countries where it is not part of the health care provisions.

## Supporting information

**S1 Table. Changes to the study protocol.** The different versions and amendments to the Study Protocol.
(DOCX)

**S2 Table. SPIRIT checklist.** Recommended items to address in a clinical trial protocol and related documents.
(DOC)

**S1 File. Protocol approved by the ethics committee.** The original protocol approved by The Regional Committee for Medical and Health Research Ethics.
(PDF)

## Acknowledgments

We acknowledge the valuable help from the therapists at the Bergen Municipality, Grieg Academy, and the Western Norway University of Applied Sciences in the running of the interventions, and the participants for entrusting us with their care.

## Author Contributions

**Conceptualization:** Tobba Therkildsen Sudmann, Christian Gold, Stefan Koelsch.

**Formal analysis:** Stavros Skouras.

**Funding acquisition:** Stefan Koelsch.

**Methodology:** Stavros Skouras, Christian Gold.

**Project administration:** Birthe K. Flo, Anna Maria Matziorinis, Tobba Therkildsen Sudmann.

**Supervision:** Stavros Skouras, Stefan Koelsch.

**Writing – original draft:** Birthe K. Flo.

**Writing – review & editing:** Anna Maria Matziorinis, Stavros Skouras, Christian Gold, Stefan Koelsch.

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
