## [Decision Letter · Decision Letter 0]

30 Mar 2022

PONE-D-22-00239Protocol: A Randomised Controlled Trial to Compare the Efficacy of Music Therapy and Physical Activity on Brain Plasticity, Depressive Symptoms, and Cognitive Decline, in a Population With and At Risk for Alzheimer’s DiseasePLOS ONE

Dear Dr. Flo,

Thank you for submitting your manuscript to PLOS ONE. After careful consideration, we feel that it has merit but does not fully meet PLOS ONE’s publication criteria as it currently stands. Therefore, we invite you to submit a revised version of the manuscript that addresses the points raised during the review process.

We look forward to receiving your revised manuscript.

Kind regards,

Burak Yulug

Academic Editor

PLOS ONE

Journal Requirements:

Reviewers' comments:

Reviewer's Responses to Questions

**Comments to the Author**

1. Does the manuscript provide a valid rationale for the proposed study, with clearly identified and justified research questions?

Reviewer #1: Yes

Reviewer #2: Yes

2. Is the protocol technically sound and planned in a manner that will lead to a meaningful outcome and allow testing the stated hypotheses?

Reviewer #1: Yes

Reviewer #2: Yes

3. Is the methodology feasible and described in sufficient detail to allow the work to be replicable?

Reviewer #1: Yes

Reviewer #2: Yes

4. Have the authors described where all data underlying the findings will be made available when the study is complete?

Reviewer #1: Yes

Reviewer #2: Yes

5. Is the manuscript presented in an intelligible fashion and written in standard English?

Reviewer #1: Yes

Reviewer #2: Yes

6. Review Comments to the Author

You may also provide optional suggestions and comments to authors that they might find helpful in planning their study.

Reviewer #1: The authors well designed a study protocol that is aimed to evaluate the pro-cognitive cognitive role of music therapy and exercise.

However, there are some minor points that would help to increase the impact and hence the relevance of the expected results of the study

1-There are no defined pulse/heart zone criteria for the exercise regimen that would be better to determine since there are strong correlations between the aeorobic exercise and heart zones (for instance, that the heart rate should be least 50% elevated than the resting state for a pro-cognitive role through various neurochemicals including also dopamine and BDNF etc.)

2-How the authors plan to discriminate the isolated effect of music therapy or exercise (it can be partly isolated ) from the general pro-cognitive effect of the social engagement (enriched environment effect?). This is an issue that should be clarified.

3-Do they plan to evaluate also some metabolic parameters which could have a role in mediating the beneficial effects. Hence, all basic and after-intervention parameters should be determined, such as glucose, and BMI, which have an isolated pro-cognitive effect if regulated properly.

Reviewer #2: General Summary: The authors present a study protocol aiming at demonstrating the effects of music therapy in a three arm RCT.

Study duration is planned for 12 months with the music therapy intervention group, an active control group (physical activity) and a passive control group (no intervention).

Theoretical background:

The study focusses on an immensely important and vulnerable patient group (AD). In my opinion the importance of this study is increased by recent developments in pharmaceutical industry reducing financial means to explore treatment options.

In general, the introduction is well written and includes up to date scientific evidence.

Methods/ design:

The three armed RCT design is sound and well planned. Inclusion criteria and exclusion criteria are well defined. Primary and secondary outcomes a clear described. The measurement of the primary outcomes includes valid MRT and test psychological methods. Power calculation using G* Power 3 is well done. The interventions are clear defined and well described.

I just like to raise three questions concerning the methods

1) the main outcome parameters Brain Age and depressives symptoms are from a clinical perspective not independent. Hence a participant with a faster

progression of the AD (= a faster progression of the Brain age gap) is more likely to develop depressive symptoms. Therefor the authors might think about how to statistically correct for this interaction

2) It is known that AD and vascular dementia often co-occur. How will the authors proceed if the inclusion criteria are met in the primary assessment, but after 12 months signs of Leukencephalopathy or similar are found?

3) The authors mention in the introduction that pharmacological interventions have limited effects in AD (And they are totally right). However psychopharmacological interventions concerning depressive symptoms are well established. Further it is known that Psychotherapy and Psychopharmacological inventions show additive effects. It might be likely that this goes as well for MT. The issue of psychopharmacology should be clearly stated in methods. Because in the worst case MT might work better but the other groups receive more Antidepressants. On how to deal with this issue valuable insight might come from Prof. Munkholm or Prof. Leucht (just a suggestion)

Conclusion:

The study protocol is sound, the study motivation well established and methods and study design are based on up to date scientific standards.-

7. PLOS authors have the option to publish the peer review history of their article (what does this mean?). If published, this will include your full peer review and any attached files.

Reviewer #1: No

Reviewer #2: **Yes: **Christian Mikutta

---

## [Author Response · Author response to Decision Letter 0]

24 May 2022

Dear editor and reviewers,

Thank you for inviting us to submit a revised version of our manuscript to PLOS ONE. We appreciate the time and effort the editor and the two reviewers dedicated to providing helpful feedback. Our responses to each point raised during the review process can be found below. We hope that our revision and responses are sufficient to meet PLOS ONE’s publication criteria. 

Academic Editor:

The heading format, figure citation, table titles, reference citation and file naming were edited to comply with the style requirements. We hopefully meet all the style requirements now. 

Our manuscript reports a study protocol without any experimental results because data collection is currently ongoing. Our Data Availability statement specifies that de-identified data will be made available upon study completion, along with the publication of the results, that will be performed through a separate article. Thereby, we believe this comment is not applicable in the case of our manuscript.

Reviewer #1:

The authors well designed a study protocol that is aimed to evaluate the pro-cognitive cognitive role of music therapy and exercise. However, there are some minor points that would help to increase the impact and hence the relevance of the expected results of the study.

1. There are no defined pulse/heart zone criteria for the exercise regimen that would be better to determine since there are strong correlations between the aeorobic exercise and heart zones (for instance, that the heart rate should be least 50% elevated than the resting state for a pro-cognitive role through various neurochemicals including also dopamine and BDNF etc.)

Response: We agree that it would be preferable to have consistent monitoring of participant’s heart zones. However, please note that the participants are observed at each session in accordance with the Borg Rating of Perceived Exertion (Borg, 1998), which has been shown to be a good estimate for heart rate during physical activity. This fact has now been added to the manuscript under the header “Description of physical activity”. Please also note that our study sample includes individuals with (early) AD, who are sometimes quite unwell (also having overall a higher mean age and low starting fitness levels), making testing physical fitness challenging, and although our study also includes younger individuals with higher fitness levels, we aimed at testing the physical fitness of all participants in the same way. 

2. How the authors plan to discriminate the isolated effect of music therapy or exercise (it can be partly isolated ) from the general pro-cognitive effect of the social engagement (enriched environment effect?). This is an issue that should be clarified.

Response: Thank you for this important point. We discuss the effect of social engagement now (in the revised discussion section). Indeed, we expect that social engagement is part of the therapeutic process which is why we made both active interventions social in nature so that potential effects will not be due to social engagement alone. As social engagement is inherent to both interventions, any variance associated to social engagement will be balanced and controlled for in group comparisons.

3. Do they plan to evaluate also some metabolic parameters which could have a role in mediating the beneficial effects. Hence, all basic and after-intervention parameters should be determined, such as glucose, and BMI, which have an isolated pro-cognitive effect if regulated properly.

Response: We agree that evaluation of metabolic parameters would have been instructive for the interpretation of the study effects. We use an extensive test battery, spreading over multiple days. Therefore, adding more measurements and doing bloodwork were originally thought to be too overwhelming for our patient groups. In terms of BMI, weight has been acquired pre and post-intervention, thus we will be able to make direction pre-/post comparisons regarding weight gain / loss in each participant group. However, we have added this important point in Section "Description of Physical Activity": "In the interest of feasibility, we did not include the assessment of metabolic parameters, nor pulse measurements during the physical activity intervention, but future studies might include such measures as they could provide further insights into their role in mediating pro-cognitive effects."

Reviewer #2:

General Summary: The authors present a study protocol aiming at demonstrating the effects of music therapy in a three arm RCT. Study duration is planned for 12 months with the music therapy intervention group, an active control group (physical activity) and a passive control group (no intervention).

Theoretical background:

The study focusses on an immensely important and vulnerable patient group (AD). In my opinion the importance of this study is increased by recent developments in pharmaceutical industry reducing financial means to explore treatment options. In general, the introduction is well written and includes up to date scientific evidence.

Methods/ design:

The three armed RCT design is sound and well planned. Inclusion criteria and exclusion criteria are well defined. Primary and secondary outcomes a clear described. The measurement of the primary outcomes includes valid MRT and test psychological methods. Power calculation using G* Power 3 is well done. The interventions are clear defined and well described.

I just like to raise three questions concerning the methods.

1. The main outcome parameters Brain Age and depressives symptoms are from a clinical perspective not independent. Hence a participant with a faster progression of the AD (= a faster progression of the Brain age gap) is more likely to develop depressive symptoms. Therefor the authors might think about how to statistically correct for this interaction.

Response: Thank you for this important question. Regarding the interaction between BrainAGE and depressive symptoms, the idea is that the BrainAGE score captures a neurological/neuroimaging "hidden" measure, whereas the depression score captures a psychiatric/behavioural "observable/visible" measure. Therefore, we want to investigate these two scores as dependent/outcome variables in two independent statistical parametric mapping models. Please note, however, that the effect of one measure alone can be assessed by controlling for the other. 

2. It is known that AD and vascular dementia often co-occur. How will the authors proceed if the inclusion criteria are met in the primary assessment, but after 12 months signs of Leukencephalopathy or similar are found?

Response: Thank you for this important point. In case of incidental findings of a non-AD severe disease, such participants would be excluded from the final statistical modelling. We have added a statement addressing this under “Statistical analyses”. Please also note that, regarding leukoencephalopathy and other conditions, ideally, we would exclude any participants that develop any severe disease, apart from AD, during the study. However, we only controlled for this with the inclusion/exclusion criteria pre-intervention because AD shows a high degree of comorbidity with multiple other health conditions. In our sample, all dementia participants had dementia diagnosed as (early) AD from the beginning and so far, no one has transitioned to dementia during the study. 

3. The authors mention in the introduction that pharmacological interventions have limited effects in AD (And they are totally right). However psychopharmacological interventions concerning depressive symptoms are well established. Further it is known that Psychotherapy and Psychopharmacological inventions show additive effects. It might be likely that this goes as well for MT. The issue of psychopharmacology should be clearly stated in methods. Because in the worst case MT might work better but the other groups receive more Antidepressants. On how to deal with this issue valuable insight might come from Prof. Munkholm or Prof. Leucht (just a suggestion)

Response: Yes, this is another important point. Our study excludes individuals with depressive disorders, therefore none of the participants without AD are on antidepressants. However, 13 of the AD participants did use antidepressants and we plan to model medication effects in a statistical analysis, in order to investigate the effect of only MT and the potential additive effect of MT combined with antidepressants. We have added this point under “Statistical analyses” in the manuscript.

Once more, we thank you for your time and insightful questions and comments, that have helped us improve the manuscript.

Sincerely, 

Birthe Kristin Flo

Corresponding author

On behalf of all authors.

---

## [Decision Letter · Decision Letter 1]

16 Jun 2022

Study protocol for the Alzheimer and Music Therapy study: an RCT to compare the efficacy of Music therapy and physical activity on brain plasticity, depressive symptoms, and cognitive decline, in a population with and at risk for Alzheimer's Disease

PONE-D-22-00239R1

Dear Dr. Flo,

We’re pleased to inform you that your manuscript has been judged scientifically suitable for publication and will be formally accepted for publication once it meets all outstanding technical requirements.

Kind regards,

Burak Yulug

Academic Editor

PLOS ONE

Additional Editor Comments (optional):

Reviewers' comments:

Reviewer's Responses to Questions

**Comments to the Author**

1. Does the manuscript provide a valid rationale for the proposed study, with clearly identified and justified research questions?

Reviewer #1: Yes

Reviewer #2: Yes

2. Is the protocol technically sound and planned in a manner that will lead to a meaningful outcome and allow testing the stated hypotheses?

Reviewer #1: Yes

Reviewer #2: Yes

3. Is the methodology feasible and described in sufficient detail to allow the work to be replicable?

Reviewer #1: Yes

Reviewer #2: Yes

4. Have the authors described where all data underlying the findings will be made available when the study is complete?

Reviewer #1: Yes

Reviewer #2: Yes

5. Is the manuscript presented in an intelligible fashion and written in standard English?

Reviewer #1: Yes

Reviewer #2: Yes

6. Review Comments to the Author

You may also provide optional suggestions and comments to authors that they might find helpful in planning their study.

Reviewer #1: The manuscript is acceptable in its current form and the authors well reponded to my major concerns.

Reviewer #2: Dear Authors,

Thank you for clarifying the issues raised in the review process.

From my siede there are nor further hindrances for publication

Congratulations for this important and well planed project

7. PLOS authors have the option to publish the peer review history of their article (what does this mean?). If published, this will include your full peer review and any attached files.

Reviewer #1: No

Reviewer #2: **Yes: **Christian Mikutta

---

## [Editor Report · Acceptance letter]

22 Jun 2022

PONE-D-22-00239R1 

Study protocol for the Alzheimer and Music Therapy study: an RCT to compare the efficacy of Music therapy and physical activity on brain plasticity, depressive symptoms, and cognitive decline, in a population with and at risk for Alzheimer's Disease 

Dear Dr. Flo:

I'm pleased to inform you that your manuscript has been deemed suitable for publication in PLOS ONE. Congratulations! Your manuscript is now with our production department. 

Kind regards, 

on behalf of

Dr. Burak Yulug 

Academic Editor

PLOS ONE